# Genome-Wide Identification of the B-Box Gene Family and Expression Analysis Suggests Their Potential Role in Photoperiod-Mediated β-Carotene Accumulation in the Endocarp of Cucumber (*Cucumis sativus* L.) Fruit

**DOI:** 10.3390/genes13040658

**Published:** 2022-04-08

**Authors:** Hesbon Ochieng Obel, Chunyan Cheng, Ying Li, Zhen Tian, Martin Kagiki Njogu, Ji Li, Qunfeng Lou, Xiaqing Yu, Zhengan Yang, Joshua Otieno Ogweno, Jinfeng Chen

**Affiliations:** 1State Key Laboratory of Crop Genetics and Germplasm Enhancement, College of Horticulture, Nanjing Agricultural University, Weigang Campus, Nanjing 210095, China; 2019204059@njau.edu.cn (H.O.O.); chunyancheng@njau.edu.cn (C.C.); 2015204016@njau.edu.cn (Z.T.); liji1981@njau.edu.cn (J.L.); qflou@njau.edu.cn (Q.L.); xqyu@njau.edu.cn (X.Y.); 2Nanjing Vegetable Science Research Institute, Nanjing 210042, China; excelee@163.com; 3Department of Plant Science, Chuka University, Chuka 60400, Kenya; martnnjogu@gmail.com; 4College of Horticulture and Landscape, Yunnan Agricultural University, Kunming 650500, China; yangzhengan@ynau.edu.cn; 5Department of Crops, Horticulture and Soil Science, Faculty of Agriculture, Egerton University, Njoro Campus, Nakuru City 20115, Kenya; jogweno@egerton.ac.ke

**Keywords:** *CsaBBXs*, transcription factors, cucumber genome, circadian signaling, carotenoid biosynthesis

## Abstract

Carotenoids are indispensable to plants and essential for human nutrition and health. Carotenoid contents are strongly influenced by light through light-responsive genes such as B-Box (BBX) genes. BBX proteins, a class of zinc-finger transcription factors, mediate many light-signaling pathways, leading to the biosynthesis of important metabolites in plants. However, the identification of the BBX gene family and expression analysis in response to photoperiod-mediated carotenoid accumulation in cucumber remains unexplored. We performed a genome-wide study and determined the expression of cucumber BBX genes (hereafter referred to as *CsaBBXs* genes) in the endocarp of Xishuangbanna cucumber fruit (a special type of cucumber accumulating a high level of β-carotene in the endocarp) using an RNA-seq analysis of plants previously subjected to two photoperiodic conditions. Here, 26 BBX family genes were identified in the cucumber genome and named serially *CsaBB**X1* through *CsaBBX26*. We characterized *CsaBBX* genes in terms of their phylogenetic relationships, exon-intron structures, cis-acting elements, and syntenic relationships with *Arabidopsis thaliana* (L.) *Heynh.* RNA-seq analysis revealed a varied expression of *CsaBBX* genes under photoperiod treatment. The analysis of *CsaBBXs* genes revealed a strong positive correlation between *CsaBBX1*7 and carotenoid biosynthetic pathway genes (*phytoene synthase*, *ζ-carotene desaturase*, *lycopene ε-cyclase*, *β**-carotene hydroxylase-1*), thus suggesting its involvement in β-carotene biosynthesis. Additionally, nine *CsaBBX* genes (*CsaBBX* 4,5,7,9,11, 13,15,17 and 22) showed a significant positive correlation with β-carotene content. The selected *CsaBBX* genes were verified by qRT-PCR and confirmed the validity of RNA-seq data. The results of this study established the genome-wide analysis of the cucumber BBX family and provide a framework for understanding their biological role in carotenoid accumulation and photoperiodic responses. Further investigations of *CsaBBX* genes are vital since they are promising candidate genes for the functional analysis of carotenoid biosynthesis and can provide genetic tools for the molecular breeding of carotenoids in plants.

## 1. Introduction

The accumulation of bioactive chemicals such as carotenoids is influenced by light-responsive genes via the regulation of transcription factors. Plant transcription factors activate or repress the transcription of their target genes to respond to several environmental impacts [1]. B-box (BBX) proteins are zinc-finger transcription factors (TF) with one or two conserved BBX domains with distinct tertiary structures stabilized by interactions with zinc ions [2]. At the N-terminus, plant BBX proteins usually include one to two conserved BBX domains (B1) and BBX 2 (B2), and some of them also have additional CONSTANS, CO-like, and TIMING of CAB1 (CCT) domains at the C-terminus [2]. According to the number of BBXs and CCTs, BBX proteins are divided into five groups. Two BBXs and a CCT make up BBXs in classes I and II, with the amino acid sequence of BBX2 varying between the two types of BBXs. Class III BBXs have one BBX1 domain and one CCT domain, whereas class IV BBXs have both B1 and B2 domains but no CCT domain. Only BBX1 is found in class V [2,3].

The BBX TF family is involved in a wide range of biological activities in plants, including seed germination [4], flowering [5,6], shade avoidance responses [7], abiotic stress responses [8], and plant phytohormone signal transduction [9]. Extensive studies on BBX genes have been carried out in *Arabidopsis thaliana* (L.) Heynh. For example, it has been reported that *AtBBX31* promotes UV-B radiation tolerance in Arabidopsis [10], while *CmBBX22* controls leaf senescence in Chrysanthemum [11]. In Arabidopsis, the heterologous expression of *CpBBX19*, a *Chimonanthus praecox* BBX transcription factor gene, improves salt and drought tolerance [12]. Additionally, heterologous BBX in apple, *MdBBX10*, improves salt and drought tolerance by regulating ABA signaling and ROS buildup [13]. In grapevine, blooming and bud dormancy are regulated by *VvCO* and *VvCOL1* (*VviBBX2* and *VviBBX5*, respectively) [14]. The previous report showed that BBX involved in photomorphogenesis (*CO/BBX1*, *BBX4*, *BBX10*, *BBX19*, *BBX20*, *BBX21*, *BBX22*, *BBX23*, *BBX24*, *BBX25*, *BBX28*, and *BBX29*) are all ubiquitinated by *constitutive photomorphogenesis 1(COP1*) and thereafter depleted in the dark by the 26S proteasome system [9,15]. Furthermore, in vitro interactions between BBX2 to 9 and BBX13 to 16 and COP1 suggest that *COP1* modulates the stability of these proteins in the dark [16]. Notwithstanding, *COP1* preferentially stabilizes *BBX11* rather than promoting its degradation [17], implying that *COP1* regulates a yet-to-be-identified *BBX11*-degrading protein. According to these studies, several BBX proteins and *COP1* and (*Elongated Hypocotyl 5* (*HY5*) are implicated in light-dependent plant development.

BBX proteins have recently been reported to play a regulatory role in secondary metabolism in fruits such as the accumulation of plant pigment anthocyanins and carotenoids. In tomato (*Solanum lycopersicum*), *SlBBX20* is a positive regulator of carotenoid accumulation [18]. By directly stimulating the production of *phytoene synthase1*, *SlBBX20* increases tomato chloroplast growth and carotenoid accumulation. Through activating *MYB10 9*, *BBX16* is a positive regulator of light-induced anthocyanin accumulation in red pears, although *PpBBX18* and *PpBBX21* are antagonistic regulators of *hypocotyl 5* in pear fruit peel [19]. In rice, *OsBBX14* induces photomorphogenesis by binding directly to the T/G-box cis-element of the *OsHY5L1* promoter under blue light conditions. In pear, *PpBBX16* (the pear homolog of *AtBBX22*) and *PpHY5* work together to trigger the expression of *PpMYB10* and other structural genes, allowing them to better control light-induced anthocyanin synthesis [20]. By combining the effects of UV exposure and a low temperature, *MdBBX20* enhances anthocyanin synthesis in apples [21]. Multiple BBXs can participate in a biological process at the same time. In *A. thaliana*, *AtBBX4*, *AtBBX21*, *AtBBX23*, *AtBBX24*, *AtBBX25*, *AtBBX31*, and *AtBBX32*, for example, control anthocyanin accumulation [10,15,22,23,24,25]. These findings show that BBX genes are involved in plant abiotic stressors, light-mediated regulations, and metabolite production. BBX genes in cucumber, on the other hand, are yet to be characterized and functionally investigated.

Cucumber (*Cucumis sativus* L.), one of the most economically important vegetable crop species, is among the first vegetable crops to have had its whole genome sequenced [26], thus offering the possibility of investigating the BBX gene family in the species. In cucumber, many gene families have been identified, including WRKY [27], MADS-box [28], Nucleotide-binding site, NBS [29], basic leucine zipper, bZIP [30], late embryogenesis abundant, LEA [31], Clvata 3/Embryo surrounding region-related, CLE [32], and so on. However, the BBX transcription factor family has not been characterized in cucumber. In this study, 26 cucumber BBX genes (hereafter referred to as *CsaBBXs* genes) were identified and characterized in cucumber through a genome-wide survey. Gene distribution, phylogenetic and synteny analyses, the architecture of exon-intron, and motif patterns were investigated. We investigated the photoperiodic effects on the expression of BBX genes and their correlation with β-carotene content and carotenoid biosynthetic genes in the endocarp of orange-fleshed Xishuangbanna cucumber. We observed *CsaBBX* genes expressing a positive correlation with β-carotene levels and associated carotenoid biosynthetic genes. This study provides a foundation for further functional studies of plant BBXs genes on responses to environmental cues and the molecular breeding of carotenoids in plants.

## 2. Materials and Methods

### 2.1. Identification and Annotation of BBX Genes in the Cucumber Genome

The Hidden Markov Model (HMM) profile for the BBX-type zinc-finger domain (PF00643) was downloaded from the Pfam database (http://pfam.xfam.org/, accessed on 2 November 2021). HMMER 3.1b2 software was employed to scan through cucumber genome (http://plants.ensemble.org/bsmart/martview/, accessed on 2 November 2021) by using this model and the Expect (e) value cutoff was 0.01. UniProt, (http://www.uniprot.org/, accessed on 2 November 2021) and NCBI Conserved Domain Search (https://www.ncbi.nlm.nih.gov/Structure/cdd/wrpsb.cgi, accessed on 2 November 2021) were used to confirm the final 26 *CsaBBX* genes that encode the BBX domain. The presence of a BBX domain was checked manually using SMART (http://smart.embl-heidelberg.de, accessed on 2 November 2021) and the Conserved Domain Database (http://www.ncbi.nlm.nih.gov/structure/cdd/wrpsb.cgi, accessed on 2 November 2021).

The various physiochemical properties of cucumber BBX proteins, such as MW, polypeptide length, pI, instability index, aliphatic index, and GRAVY, were investigated using the ExPASy online tool (http://web.expasy.org/protparam/, accessed on 4 November 2021). To estimate the subcellular localization of cucumber BBX proteins, we used WoLF PSORT online software (http://www.genscript.com/psort/wolf_psort.html, accessed on 4 November 2021). Open Reading Frame (ORF) numbers were calculated using the NCBI website (https://www.ncbi.nlm.nih.gov/orffinder/, accessed on 4 November 2021).

### 2.2. Alignment and Phylogenetic Analysis

Conserved domain alignment analysis of amino acid sequences of the BBX and CCT domains, which were aligned within MEGA and consecutively imported into GeneDoc 2.6 for shading, was conducted and sequence logos were created using Web logo 3 (http://weblogo.threeplusone.com, accessed on 8 November 2021). The Muscle module within the MEGA 7.0 software package was used to align a sequence of full-length proteins, and phylogenetic trees were constructed by utilizing the Neighbor-Joining (NJ) approach with 1000 bootstrap replications, and the following parameters: Poisson model, uniform rates, same (homogeneous), and pairwise deletion. BBX protein sequences from *Arabidopsis thaliana* (*AtBBX*), *Zea mays* (*ZmaBBX*), *Solanum lycopersicum* (*SlyBBX*) and *Cucumis melo* (*CmeBBX*) were downloaded from genome databases maintained for each species.

### 2.3. Analysis of Gene Structures and Conserved Motifs of CsaBBXs 

Details of the *CsaBBX* gene structure, including the intron and exons, were obtained from the cucumber genome annotation (http://cucurbitgenomics.org/organism/2/, accessed on 10 November 2021) and visualized using Gene Structure Server version 2.0 (http://gsds.gao-lab.org/index.php, accessed on 10 November 2021). Conserved motifs of all BBX proteins were identified using the online MEME analysis tool (http://meme-suite.org/tools/meme, accessed on 10 November 2021) with the maximum number of motifs being set at 15, and other default parameters. Only motifs with E-value < 0.05 were present. TBtools software (version 1.098696). was used to draw the map of the conserved motif.

### 2.4. Chromosomal Locations, Patterns of Gene Duplication, and Synteny Analysis

The chromosomal locations of each *CsaBBX* gene were identified according to physical location information from the cucumber genome annotation (http://cucurbitgenomics.org/organism/2, accessed on 12 November 2021). Syntenic blocks for the cucumber BBX genes and between cucumber and Arabidopsis were identified and analyzed using the MCScanX in Tbtools. Synteny analysis and chromosomal location diagrams were generated in a globe plot using the program Circos-(http://circos.ca, accessed on 12 November 2021) in the Tbtools. The non-synonymous (Ka) and synonymous (Ks) substitution rates of each gene pair were calculated using the TBtools [33]. The Ks values were used to calculate the divergence time with the following formula: T = Ks/2λ (λ = 6.56 × 10^−9^ for cucumber) [34].

### 2.5. Cis-Acting Element Analysis in the Promoter of CsaBBXs

For cis-acting element analysis, the 1500 bp upstream transcriptional start site of each *CsaBBX* gene was examined using Plant Care (http://bioinformatics.psb.ugent.be/webtools/plantcare/html/, accessed on 12 November 2021) [35]. 

### 2.6. Gene Ontology (GO) and Functional Annotation

The functional characterization and annotation of *CsaBBX* sequences were performed using Pannzer rapid functional annotation server (http://ekhidna2.biocenter.helsinki.fi/sanspanz/, accessed on 25 November 2021) with default parameters. The functional categories were visualized using the bar chart generator tool (https://graphmaker.imageonline.co/barchart.php, accessed on 25 November 2021).

### 2.7. Plant Materials, Photoperiod Treatment, and Expression of CsaBBX

Plant materials: Xishuangbanna cucumber (XIS) (*Cucumis Sativus* L. var. Xishuangbannesis Qi et Yuan), a semi-wild species that is orange-fleshed, signifying abundant β-carotene. β-carotene biosynthesis is influenced by environmental factors, including photoperiod signaling. *BBX* has photomorphogenic effects and subsequently affects plant metabolites. 

The expression patterns of the *CsaBBX* genes in *C. s*. var. Xishuangbannesis in the fruit endocarp during different developmental stages under photoperiod inductions were analyzed based on our previous raw RNA-seq reads deposited in NCBI Sequence Read Archive (SRA) under accession PRJNA782229 in the SRA database (https://www.ncbi.nlm.nih.gov/sra/PRJNA782229, accessed on 1 December 2021). Briefly, A highly inbred “XIS” cucumber line known as South-West China cucumber (SWCC8) was used for RNA-Seq. The SWCC8 is an inbred line derived from a natural “XIS” cucumber collected from Yunnan Province in China (21°09′ to 22°36′N altitude, and 99°58′ to 101°50′E longitude, elevation 800–1200 m) [36]. SWCC8 has been identified as an SD plant and is therefore sensitive to the photoperiod [37,38]. Crop establishment and photoperiod treatment were conducted as outlined in our previous article [38] with minor modifications. Artificial fluorescence tubes with white light were used for photoperiod adjustments each with a different photoperiod treatment, SD (8 h/16 h, day/night) and equal-day (ED, 12 h/12 h, day/night), whereas all other environmental factors were the same; that is, the temperature, relative humidity, and light intensity were maintained at 27 °C/20 °C ± 2 (day/night), 80% relative humidity, and 800 μmol·m^−2^·s^−1^ photon flux density. The experiment consisted of three replications per treatment, with 8 plants established in pots filled with peat: vermiculite of 3:1 ratio per treatment. Plants were maintained under these treatments throughout the growth period. Routine management such as watering, nutrient supply, pest, and disease control was carried out on a need basis. Flesh fruit samples were collected from the endocarp of SWCC8 fruit at the sequential fruit developmental phase after manual pollination under SD and ED conditions. The fruit samples were harvested at 10, 20, 30, and 40 days after pollination (DAP). Each sample consisted of three biological replicates collected from three individual plants. Materials were immediately placed in liquid nitrogen for about 1 min and stored at −80 °C until use. Total RNAs were extracted using the TRIzol reagent (Invitrogen, Carlsbad, CA, USA). A total of 5 μg of RNA per sample was used as input material for the RNA sample preparations. cDNA libraries were generated using a NEBNext^®^UltraTM RNA Library Prep Kit for Illumina^®^ (NEB, CA, USA) following the manufacturer’s protocol. The library was diluted to 1.5 ng/μL to check the distribution of insert size using an Agilent Bioanalyzer 2100 system version B. 02.11 (Agilent Technologies, Inc., Santa Clara, CA, USA). Real-time quantitative PCR (RT qPCR) was applied to accurately determine the concentration of the library (concentration > 2 nmol was the effective concentration of the library). The clustering of the index-coded samples was performed on a cBot Cluster Generation System using a TruSeq PE Cluster Kit v4-cBot-HS (Illumina) according to the manufacturer’s instructions. After cluster generation, the library preparations were sequenced on an Illumina Hiseq 2500 platform and paired-end reads were generated. 

The raw reads library was mapped to the reference genome sequence http://cucurbitgenomics.org/organism/20, accessed on 3 June 2021) using TopHat, and the expression levels were normalized to Fragments Per Kilobase Million (FPKM) using Cufflinks. Genes with Pfdr (adjusted *p*-value) < 0.05 were defined with a significantly different expression. We cross-referenced the gene IDs of the CsaBBX of version 2.0 with those of the cucumber genome v3.0. Gene expression analyses were visualized in the form of a heatmap using Tbtools. 

### 2.8. β-Carotene Extraction and HPLC Analysis

β-carotene was extracted according to the methods described by Alagoz et al. [39]. Briefly, fruit flesh portions (endocarp) were homogenized under liquid nitrogen, and 0.5 g samples of fruit flesh powder were placed into a 2 mL centrifuge tube. The samples were extracted with 500 μL of acetone-ethyl acetate (EtOAc) (6:4 *v*/*v*) containing 1% butylated hydroxytoluene (BHT). The samples were centrifuged (Eppendorf centrifuge 5810R New Brunswick, USA) at 4 °C for 10 min at 12,000 rpm. The upper ethyl acetate phase (400 μL) was then transferred to new 2 mL microcentrifuge tubes and centrifuged for 5 min under the same revolutions. The 300 ul of ethyl acetate extracts was dried under a gentle stream of nitrogen, dissolved in 1 mL of acetone, and then stored at −20 °C until HPLC analysis. HPLC analysis of sample extract was carried out using a Waters Alliance 2695 system (Waters Corp., Milford, MA, USA). Chromatography was carried out at 25 °C with an elution program described by Alagoz et al. [39]. β-carotene was identified based on retention time. β-carotene concentrations were calculated by converting peak areas to molar concentrations by comparison with carotenoid standards of known concentration run on an HPLC. The β-carotene standard was purchased from Sigma-Aldrich (Sigma-Aldrich Co. Ltd., Shanghai, China). The β-carotene was quantified as μg/g fresh weight (fw). 

### 2.9. Quantitative RT-PCR Analysis 

In this experiment, *10 CsaBBX* genes were randomly selected to test for *CsaBBX* genes involved in the endocarp of “XIS” cucumber under photoperiodic treatments. An RNA kit (Takara, Dalian, China) was used to extract total RNA from the isolated endocarp tissues according to the manufacturer’s instructions. Nanodrop ND-spectrophotometer (NanoDrop Technologies, MA, USA) was used to measure the quality and concentration of each RNA sample. The Prime Script RT reagent kit (Takara, Dalian, China) was used to reverse transcribe RNA (RNA (1 µg treated with DNaseI) into cDNA. The Primer 5.0 software was used to design specific primers according to the *CsaBBX* gene sequences. The cucumber β-actin gene (ID number: *Csa2G301530*) was used as the internal control to normalize the expression level of the target genes among different samples [38]. Three biological and three technical replicates were used. RT-qPCR reactions were performed in 20 μL volume including 10 μL of Hieff qPCR SYBR^®^ Green Master Mix (Yeason, Shanghai, China), 2 μL of cDNA, 0.4 μL of each primer, and 7.2 μL of sterile deionized water on Bio-Rad CFX 96 fluorescence quantitative PCR platform (Bio-Rad, CA, USA). The reaction mixture was first pre-degenerated at 95 °C for 5 min, followed by 40 cycles of denaturation at 95 °C for 10 s and annealing at 60 °C for 30 s. The temperature increase increased from 65 to 95 °C at a rate of 0.5 °C every 10 s to verify the amplification specificity. The comparative Ct value method was employed to analyze the relative gene expression level. The RNA level was expressed relative to the actin gene expression level following the 2^−∆∆CT^ method [40]. Relative quantitative data were calculated according to the σ method: normalization (ΔCT = CT (sample) − CT(ACTIN)); ΔΔCT = ΔCT (sample A) − ΔCT (sample B). The qRT-PCR data were analyzed with SPSS v.21.0 software for variance analysis, and the Waller–Duncan (W) method was used for comparison at the *p* < 0.05 level. The gene-specific primer pairs for RT-qPCR were designed from unique gene regions using primer premier 5.0 and listed in (Appendix A). 

## 3. Results

### 3.1. Identification and Characterization of Cucumber BBX Genes

To identify and obtain the BBX genes in the cucumber genome, the hidden Markov model (HMM) profile of the BBX domain (Pfam00643) was employed to perform a global search of the cucumber genome. After analyzing the conserved domain and removing the redundant sequences, a total of 26 putative *CsaBBX* genes were identified in cucumber. The protein sequences of *CsaBBX* protein are provided in the Appendix A. For the sake of nomenclature and consistency, these *CsaBBX* genes were serially named as *CsaBBX1* through *CsaBBX26*. The detailed information of *CsaBBXs* was listed in Table 1, including gene name, protein length, chromosome location, molecular weight, theoretical isoelectric point, aliphatic index, and grand average of hydropathicity (GRAVY). The 26 *CsaBBX* proteins had diverse molecular lengths and weights, ranging from 123 (*CsaBBX23*) to 542 (*CsaBBX20*) in amino acid length. *CsaBBX22* had the lowest molecular weight value (13.12 kDa), while the highest molecular weight (60.05 kDa) was observed for *CsaBBX20*. The theoretical isoelectric points (PI) of these *CsaBBX* proteins varied from 4.31 (*CsaBBX11*) to 8.4 (*CsaBBX10*), with an average of 6.02 and the value of the aliphatic index ranging from 53.72 (*CsaBBX11*) to 86.66 (*CsaBBX26*). The GRAVY of all *CsaBBXs* was less than zero, indicating the hydrophilic nature of *CsaBBX* proteins (Table 1). The majority of *CsaBBX* (13 *CsaBBXs*) proteins were predicted to be located on the nucleus by WoLF PSORT, but a few of them may be located in other subcellular compartments, such as chloroplasts and the cytoplasm (Table 1).

### 3.2. Phylogeny, Protein Structure, and Conserved Domains of the Cucumber BBX Proteins 

The 26 cucumber BBX proteins were classified into five structural groups based on amino acid sequence conservation, the presence or absence of numerous BBX domains, and the presence or absence of the CCT domain (Figure 1). Two BBX domains and one CCT domain were found in Group I, which included four proteins. Group II (four members) proteins were distinguished from Group I proteins; however, the BBX 2 in Group II seemingly had the same sequences as BOX as observed by the same color pattern. Group III comprises one BBX domain and one CCT domain, which included four proteins. Group IV proteins (10 proteins) had two BBX domains. However, *CsaBBX26* had double BBX 2. Group V proteins (four proteins) only had one BBX1. Twelve of the 26 *CsaBBX* genes contain the CCT domain, thus suggesting their active role in photomorphogenesis and flower development.

A total of 145 BBX proteins (including 26 from cucumber, 32 from Arabidopsis, 31 from tomato, 31 from maize and 25 from melon) were utilized to build a phylogenetic tree to examine the evolutionary relationship and likely functional divergence of the *CsaBBX* gene family (Figure 2, Appendix A). This allowed us to categorize the cucumber *CsaBBX* proteins into five clades, most of which are related to their structural groups, and are in line with prior findings in Arabidopsis (2). In the above clades/groups, however, certain exceptions were found. The structural group placed the *CsaBBX2* protein in Group V; however, it was found in phylogenetic clade IV (Figure 1). The majority of the *CsaBBXs* clustered together with proteins from Arabidopsis, melon, and tomato, rather than maize, which is consistent with cucumber’s closer kinship to the three eudicots and being in a similar family to the melon.

The conserved sequences of the domains comprise C-X2-C-X7–8-C-X2-D-X-A-XL-C-X2-C-D-X3-H and C-X2-D-X-A-X-L-C-X2-C-D-X3-H-X-A-N-X—L-X3-H for the BBX1 and BBX2 zinc-finger domains, respectively. In addition, the CCT domains of twelve grapevine proteins were highly conserved, having the form R-X5-R-Y-X2-K-X3-R-X3-K-X2-R-Y-X2-R-KX2-A-X2-R-X2-R-X2-G-R-F-X-K. The conservation of amino acids with these motifs is depicted graphically in (Figure 3A). Because of five conserved amino acid residues (two Asps, Ala, Leu, and Asn) in all B-box1 domains, the alignment of the protein sequences revealed that now the B-box1 domain was more conserved than that of BBX2 (Figure 3B).

### 3.3. Analysis of Phylogeny, Exon–Intron Structures, and Conserved Protein Motifs of CsaBBX Genes 

We investigated the conserved protein motifs encoded by the genes, as well as exon-intron architectures, to gain a better understanding of the *CsaBBX* gene family’s conservation and diversification (Figure 4). The intron/exon architectures were studied by aligning the cDNA sequences and corresponding genomic DNA sequences to gain a better understanding of the evolutionary links of the *CsaBBX* genes (Figure 4B). *CsaBBX2*, *CsaBBX6*, *CsaBBX10*, *CsaBBX22*, and *CsaBBX25* each had only one exon, while *CsaBBX20* had the most exons (6). *CsaBBX20* had six exons, while the other three members of the Group I had four. Furthermore, the highly comparable gene structure was discovered in the same group of *CsaBBX* genes, with slight differences. For example, except for *CsaBBX20* (six exons), all three *CsaBBXs* in Group I had four exons, while all members of Group II had two exons. The number of exons in members of Group IV ranged from one to five. It’s also worth noting that none of the members of Group V (except *CsaBBX 11*) and Group IV’s *CsaBBX6* have introns. These findings indicated that exon gain or loss happened during the evolution of the *CsaBBX* gene family and that gene structure reveals the *CsaBBXs’* evolutionary link.

Further, we analyzed the motif structure of *CsaBBX* genes (Figure 4C). Multilayer consensus sequences E-values, as well as fifteen conserved motifs, are shown in Appendix A. The results revealed that, in terms of width, motifs 2 and 8 were the largest (50 each), followed by motif 15 and then motif 4, with motif 4 being the smallest. BBX1 (motif 1), BBX2 (motif 3), and CCT were all assigned to three motifs (motif 2). In all of the *CsaBBXs*, motif 1 was found (Figure 4C). Groups I, II, and III of *CsaBBXs* have a distinct motif 2. Only Group II *CsaBBX* genes contained motif 5. Interestingly, motifs 8, 9, and 15 were found only in Group III, Group IV, and Group I, respectively, suggesting that functional divergence of BBX genes is a factor. Motifs 6 and 7 were found in all Group I and II members except *CsaBBX8* in Group 1, but they were also found in *CsaBBX17* in Group IV, implying that *CsaBBX17* developed from a Group II gene. Our findings revealed that the most closely related individuals in the evolutionary tree shared common motifs based on alignment and location, implying that they may share a biological role.

### 3.4. Chromosomal Location, Homologous Gene Pairs, and Synteny Analysis

The 26 *CsaBBX* genes were discovered to be widely dispersed among cucumber chromosomes based on their identified genomic positions (Figure 5A). *CsaBBX* genes were found on chromosome 2 (six), chromosome 6 (five), and chromosomes 1, 4, and 7 (four). One and two *CsaBBX* genes were found on chromosomes 3 and 5. Tandem and segmental duplications were widespread during the evolution and extension of the gene family. Tandem duplication usually results in gene clusters, but segmental duplication might cause family members to become separated [41]. On the cucumber chromosome, only one tandem duplication cluster (*CsaBBX3/CsaBBX4*) in the *CsaBBX* gene family was found (5B). The cucumber genome then revealed six pairs of duplicated segments (CsaBBX1/CsaBBX8, *CsaBBX6/CsaBBX23*, *CsaBBX20/VvBBX24*, *CsaBBX14/CsaBBX19*, *CsaBBX11/VvBBX22*, and *CsaBBX13/CsaBBX21)* in the *CsaBBX* gene family (Figure 5B). The findings suggested that segmental duplication events may have had a larger role in the growth of the *CsaBBX* gene family in cucumber than tandem duplication. In addition, we determined the value of non-synonymous (Ka) versus synonymous (Ks) substitution rates (Ka/Ks) for the tandem and segmentally duplicated gene pairs which can be used as a proxy for a gene’s selection pressure during evolution. Our findings revealed that all of the Ka/Ks values were smaller than 1, except for *CsaBBX13/CsaBBX21*, implying that the *CsaBBX* genes evolved largely under purifying selection The most recent (*CsabbX3/CsaBBX4*) and oldest (*CsaBBX6/CsaBBX23*) duplication occurrences took place around 27 and 130 million years ago (Mya), respectively (Appendix A). Further, we looked into the orthologous BBX gene pairs found in cucumber and Arabidopsis (Figure 5C). According to the findings, 18 cucumber *CsaBBX* genes and 16 Arabidopsis *CsaBBX* genes are orthologous gene pairs, resulting in 27 syntenic links between the two species (Appendix A).

### 3.5. Identification of Cis-Elements in the Promoters of CsaBBX Genes

To better understand the transcriptional regulation and the gene function of *CsaBBX*, the cis-elements in the promoter regions of the *CsaBBX* (1.5 kb of genomic DNA sequence upstream of the translation start site) were used to search the database (Figure 6: Appendix A). CAAT-box and TATA-box, the conventional promoter elements, were found in all the *CsaBBX* promoters. A series of cis-elements involved in plant growth and development, phytohormone responses, and stress responses were identified (Figure 6A; Appendix A). The light responsiveness cis-regulatory elements account for the largest proportion (up to 49%) in the total across the promoters of 26 *CsaBBX* genes followed by defense and stress-related elements (22%) (Figure 4B).

The light responsiveness elements contain different kinds of *cis*-regulatory elements such as G-box, ACE, Box 4, TCCC-motif, GTI-motif, GATA-motif, circadian, and TCT-motif MRE, among others (Figure 6B and Appendix A). Additionally, the *cis*-regulatory elements associated with hormone responses (including auxin auxin-responsive element (Aux-RR-core and TGA-element), salicylic acid (TCA-element), gibberellins (GARE-motif, P-box, and TATC-box), abscisic acid (ABRE), MeJA (CGTCA-motif and TGACG-motif), and ethylene-responsive element (ERE)), stress response (to drought, low temperature, defense, and stress), meristem expression, and anaerobic induction were also identified in promoter sequences of the cucumber (Figure 6C). In addition, as shown in Figure 6A, the CAT-box involved in meristem expression, the zein metabolism regulation element (O2 site), endosperm expression regulation element (GCN4_motif) the palisade mesophyll, and flavonoid biosynthetic were also found in the promoters of the *CsaBBX* genes. Our findings imply that the cis-element-containing promoter regions of *CsaBBX* genes were important in the light and stress responses.

### 3.6. Gene Ontology Annotation Analysis of CsaBBX Genes

The potential functions of *CsaBBXs* in cucumber were predicted by gene ontology (GO) annotation analysis. In this study, the 26 *CsaBBX* genes were assigned to different GO categories. The results showed that the 26 *CsaBBXs* genes participated in the biological process (16 *CsaBBX* members), molecular function (7), and cellular component (2) (Figure 7). The “regulation of transcription DNA-templated’ (14 *CsaBBXs*) was the dominant group in the biological process category, comprising 16.92%, followed by photomorphogenesis (12.8%) and the positive regulation of photomorphogenesis (8.5%). In the molecular function, “Zinc ion binding (43%) and Cis-regulatory binding (23%)” form the major components. Regarding the cellular component category, only two components were identified with the nucleus (73%), “forming the largest portion” and an “integral component of membrane”, accounting for 23%. Additional details of each *CsaBBX* gene enrichment are provided (Appendix A).

### 3.7. Potential CsaBBX Proteins Involved in β-Carotene Biosynthesis in the Endocarp of Orange-Fleshed Xishuangbanna Cucumber Fruits and Their Expression Profiles under Photoperiod Treatment

The flesh color intensity of “XIS” fruit endocarp gradually changes as fruit matures and becomes more orange at 40 days after anthesis (Appendix A). Our analysis of β-carotene constantly showed significantly higher levels under day-neutral compared to short-day (Appendix A). The highest peak under day-neutral was 8 μg/g fw compared to 5 μg/g fw under short-day conditions. This result signifies that carotenoid accumulation in the endocarp of “XIS” cucumber is light-driven.

BBX genes are affected by the circadian clock, a light-dependent diurnal oscillation, and consequently affect plant metabolites [19]. To test this, the expression patterns of the *CsaBBX* gene were further examined based on the previous transcriptomic data (SRA PRJNA782229). The expression levels of most genes in the endocarp of “XIS” cucumber showed significant differences. These results indicated that under short-day conditions, *CsaBBX1*, *CsaBBX6*, *CsaBBX7*, *CsaBBX8*, *CsaBBX9*, *CsaBBX11*, *CsaBBX12*, *CsaBBX14*, *CsaBBX16*, *CsaBBX18*, *CsaBBX20*, and *CsaBBX25* were up-regulated at different developmental stages. Under equal-daylight conditions, *CsaBBX1*, *CsaBBX4*, *CsaBBX5*, *CsaBBX6*, *CsaBBX11*, *CsaBBX13*, *CsaBBX15*, *CsaBBX16*, *CsaBBX17*, *CsaBBX18*, *CsaBBX19*, and *CsaBBX21* were up-regulated, mainly at early fruit growth stage (10 DAP) (Figure 8). We performed a correlation analysis of *CsaBBX* genes and carotenoid levels in “XIS” cucumber. The findings indicated that the expressions of *CsaBBX9*, *CsaBBX12*, *CsaBBX 18*, *CsaBBX20*, *CsaBBX22*, *CsaBBX23*, and *CsaBBX25* were significantly (*p* < 0.05) correlated with both β-carotene content. Therefore, these genes might participate in carotenoid biosynthesis through a light-signaling mechanism due to the presence of light-responsive cis-elements in the promoter sequences (Appendix A). 

Previous studies suggested that some BBX proteins are involved in carotenoid biosynthesis [18]. To further identify the *CsaBBX* gene which may be involved in carotenoid synthesis, we analyzed the gene co-expression between the *CsaBBX* gene and carotenoid-related genes based on our transcriptomic data (SRA PRJNA782229) of “XIS” Xishuangbanna cucumber fruits (Figure 9). The results showed a positive correlation of *CsaBBX17* with carotenoid biosynthetic genes, *phytoene synthase* (*CsaPSY*), *ζ-carotene desaturase* (*CsaZDS*), *ζ-carotene isomerase* (*CsaIZO*), *lycopene ε-cyclase* (*CsaLYCB*), and β-*carotene hydroxylase-1* (*CsaBCH1*) under day-neutral conditions while showing a significant negative correlation with carotenoid cleavage genes, *zeaxanthin epoxidase* (*CsaZEP*), *CCD*, *carotenoid cleavage dioxygenase* (*CsaCCD*), *9-cis**-epoxycarotenoid dioxygenase* (*CsaNCED*), *abscisic acid-8 hydroxylase* (*CsaABA 8*) and with all carotenoid genes under short-day conditions. Under the short-day condition, *CsaBBX18* and *CsaBBX20* showed a positive correlation with *CsaPSY*, while *CsaBBX2* had a positive correlation with (*CsaCBH1*). The results suggest a diverse regulation of carotenoid-related genes by BBX genes under different light conditions. We predicted that *CsaBBX17* participation in β-carotene biosynthesis is dependent on the presence of long light duration. This gene can be selected as a candidate gene for further study to determine its functional role in carotenoid biosynthesis since its upregulation is concurrent with the associated carotenoid biosynthetic genes and carotenoid accumulation.

### 3.8. qRT-PCR Analysis of Selected CsaBBX Genes Involved in Carotenoid Differentiation in the Endocarp of “XIS” Fruit under Photoperiodic Conditions

Ten genes were selected for qRT-PCR expression analysis (Figure 10). The genes selected possess light-responsive cis-elements or circadian cis-element. The expression level of these 10 *CsaBBX* genes in the endocarp fruit of orange-fleshed “XIS” cucumber under 8L/16D vs. 12L/12D at four fruit development stages were analyzed by qRT-PCR. *CsaBBX 4*, *CsaBBX5*, *CsaBBX9*, *CsaBBX11*, *CsaBBX13*, *CsaBBX15*, *CsaBBX17*, *CsaBBX22*, and *CsaBBX25* tended to show significantly higher expression levels at most of the data collection points under equal light conditions than under short-day conditions. *CsaBBX23*, however, showed higher expression levels under short-day conditions than under equal-day conditions. The results obtained by qRT-PCR are in tandem with those presented in Figure 8 above.

## 4. Discussion

The B-box gene family has been characterized in a variety of plant species, including *Arabidopsis thaliana* [15], *Oryza sativa* [42], *Arachis duranensis* [43], *Malus domestica* [44], *Pyrus bretschneideri* Rehd [45], *Vitis vinifera* [14], *Solanum lycopersicum* [46], and *Brassica rapa* [47], among others. Although the cucumber vegetable crop’s full genome has been sequenced, the BBX gene family is yet to be identified by a genome-wide study in cucumber. This is the first time that bioinformatics technologies have been used to identify and define the BBX gene family in cucumber.

### 4.1. Characteristics of the Cucumber BBX Gene Family

A total of 26 *CsaBBX* genes were identified and classified into five clades (Groups I-V) in cucumber (Table 1). Group IV harbored the largest number of BBX genes (Figure 1). The analysis of the exon/intron structure and conserved motifs revealed that the BBX genes in each subfamily have special characteristics. Based on comparisons of gene structures, the number of exons/introns and motifs vary within subfamilies, although they tend to be comparable within each subfamily with slight variations (Figure 4). Gene duplication events such as tandem, segment, and transposition duplications are significant in genomic rearrangement, which frequently leads to the growth of gene families [41]. In cucumber, one tandem duplication and six segmental duplications were discovered in the *CsaBBX* genes (Figure 5B), implying that the BBX genes have expanded to a vast extent. Most BBX genes in cucumber may have undergone an early divergence or be obtained through gene transposition, which is consistent with previous findings that cucumber has few tandems due to a lack of recent whole-genome duplication [26]. The number of *CsaBBXs* in cucumber is lower than that in other plants, such as Arabidopsis (thirty-two genes) [2], rice (thirty genes) [42], tomato (twenty-nine genes) [46], pears (thirty-seven genes), [48], potato (thirty genes) [49], and apple (sixty-four genes) [44], but is higher than that in pepper (twenty-four genes) [8], peach (twenty-two) [50], diploid wild strawberry (*Fragaria vesca*; sixteen [51]) and sweet cherry (*Prunus avium*) (fifteen) [52]. 

Our phylogenetic analysis revealed that the *CsaBBXs* in cucumber can be classified into five groups and possess BBX1, BBX2, and CCT conserved domains, which were similar to those in other species [2,42,44,46,48,49]. The results of the motif analysis of *CsaBBX* proteins are consistent with those of domain analysis. The gene structures (UTR, intron, and exon) of *CsaBBX* genes of the same group have an equal number of exons and introns, with a few exceptions observed in some groups, particularly in Group IV, in which exons range from 1 to 6, and members of Group V except for *CsaBBX11* lacked introns. In all *CsaBBX* promoter regions, a series of cis-acting elements were predicted (Figure 6 and Appendix A), four of which were found in all *CsaBBX* promoters, and the dominant cis-acting elements are comprised of light-responsive elements. This finding concurs with a prior study, which found that BBX genes play a role in photomorphogenesis [53]. Each *CsaBBX* promoter region had one or more G-boxes. By attaching the G-box to the promoter region of *CsaBBX* genes, *CsaBBX* proteins may regulate each other. The cis-elements involved in circadian rhythm were found in the promoter regions of *CsaBBXs* (Appendix A). These findings are similar to those found for Arabidopsis, where *AtBBXs* are controlled by the circadian clock and some of them participate in light-related processes such as flowering, shade avoidance, and photomorphogenesis [15]. In the *CsaBBX* gene promoter regions, cis-elements connected with numerous hormones, biotic/abiotic stressors, and growth and development-related activities were also predicted. These data imply that different signaling mechanisms can influence *CsaBBX* expression. Go ontology analysis further indicates that most *CsaBBX* genes are involved in photomorphogenic biological processes and zinc ion binding activity, and are mainly localized in the nucleus.

### 4.2. Potential Role of CsaBBX Genes in β-Carotene Biosynthesis in the Endocarp of “XIS” Cucumber and Photoperiod Response

Carotenoids provide health benefits, and higher amounts of carotenoids in crop species can boost nutritional and aesthetic quality. Light is an important environmental factor that regulates the development and coloration of fruits. Therefore, the study of *CsaBBX* as an important light signal regulator is expected to be an interesting entry point for exploring the relationship between light and the appearance and the intrinsic quality of “XIS” cucumber. Several BBX genes have been linked to light control in the past. Light-mediated development is regulated by *AtBBX21* and *AtBBX22* in *A. thaliana* [25]. Additionally, in grapevines, light exposure up-regulated *VvBBX36*, *VvBBX43*, *VvBBX44*, *VvBBX47*, and *VvBBX50* in “Jingxiu” and “Muscat Hamburg” compared to light exclusion. 

Here, the effect of a photoperiod treatment and its association with β-carotene accumulation was examined. As shown in Figure 8, the expression levels of 11 *CsaBBXs* including (*CsaBBX1*, 4, 5, 6, 11, 13, 15, 16, 17, 18, and 19 were up-regulated under equal-day exclusion in “XIS” cucumber under different conditions. Many *CsaBBX* genes are regulated by light-signaling pathway components such as phytochromes and photomorphogenesis downstream regulators such as HY5, long after far-red light 1, Far-red impaired response 1 and phytochrome 3), as evidenced by their enhanced expression throughout the long-day phase [49]. Other *CsaBBX* genes are specifically expressed during the short-day phase, indicating that they could be regulated by dark-specific factors such as *Phytochrome*-*Interacting Factors* (PIFs) which collaborate with phytochromes in the control of gene expression. The *CsaBBX* in cucumber demonstrated significant changes in transcriptional levels and possible roles due to variances in chromosome position, amino acid sequence, and promoter sequences. Our findings imply that BBX proteins are involved in several developmental processes that are tightly regulated by the diurnal cycle’s light and dark phases. The fact that members of distinct structural groups represent the light- and dark-specific expression groups suggests that all BBX types are involved in these activities. *BBX21*, a two-BBX domain protein, can bind to a T/G-box in the *HY5* promoter, stimulating its expression and influencing g photomorphogenesis [22].

BBX genes also play critical roles in carotenoid biosynthesis in plants. The previous report indicates that *SlBBX20* can boost the expression of *Phytoene synthase 1* (*PSY1*), which encodes a critical enzyme in carotenoid biosynthesis, by unswervingly binding to a G-box motif in its promoter [18]. Increased expression levels of *PSY1* result in increased amounts of carotenoids in *SlBBX20* overexpression lines. This indicates that BBX genes participate in carotenoid biosynthesis through the promoter motifs. In our study, seven *CsaBBXs* (*CsaBBX9*, *CsaBBX12*, *CsaBBX18*, *CsaBBX20*, *CsaBBX22*, *CsaBBX23*, and *CsaBBX25*) were significantly (*p* < 0.05) correlated with both β-carotene and total carotenoid levels. *CsaBBX9*, *12*, *18*, and *25* equally possess the G-box motif that could promote their activity for carotenoid biosynthesis. In our study, a co-expression analysis suggests that *CsaBBX17* positively correlates with carotenoid biosynthesis genes (*PSY1*, *IZO*, *ZDS*, *LYCB*, *CHYB1*) under long light days, thus suggesting its role in carotenoid biosynthesis under the light phase compared to short day or darkness. The light-responsive elements may help regulate *CsaBBX* gene expression in the light, whereas the B- and E-box variations of the G-box elements may help regulate *CsaBBX* gene expression in the dark. Testing the selected *CsaBBX* genes by qRT-PCR in “XIS” cucumber fruit endocarp subjected to two photoperiod regimes (Figure 10) corroborated the results of RNA-seq data in Figure 8. Therefore, these *CsaBBX* genes (those confirmed by qRT-PCR, correlation analysis with β-carotene content and co-expression analysis with carotenoid genes) may be associated with the accumulation of β-carotene in the endocarp of “XIS” fruit. Due to the complexity of the BBX gene function, all the candidate *CsaBBX* genes need further analysis to verify their functions in the regulation of carotenoid biosynthesis.

## 5. Conclusions

In this study, 26 *CsaBBX* genes were identified in cucumber, and the *CsaBBX* gene family was analyzed in detail, including conserved domains, phylogenetic relationships, gene structure, chromosome placement, gene duplication, cis-acting elements, and expression pattern analysis. The *CsaBBX* promoter sequences contained numerous cis-acting elements, indicating that *CsaBBX* genes are involved in complex regulatory mechanisms that control the development of and responses to abiotic stresses. The expression analysis revealed that *CsaBBX* genes might play important roles in fruit development and carotenoid biosynthesis by modulating multiple signaling pathways. The genome-wide investigation of the *CsaBBX* gene provides a good framework for functional assessments of BBX genes in cucumber. Our findings also give important information by identifying putative cucumber BBX genes that might be involved in photoperiod-driven β-carotene accumulation in the endocarp of “XIS” cucumber fruit. This study not only provided a scientific foundation for the comprehensive understanding of the cucumber BBX gene family but was also helpful for screening more candidate genes for breeding new varieties with photoperiodic adaptability and improved nutritional value. 

## Figures and Tables

**Figure 1 genes-13-00658-f001:**
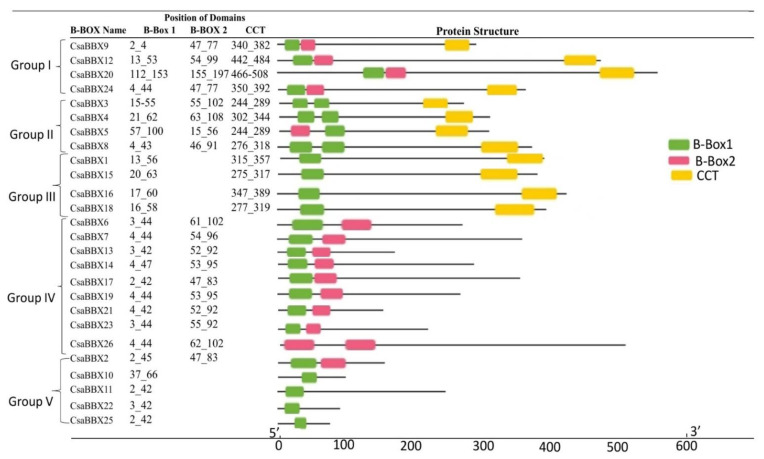
Structures of the *CsaBBX* proteins with the composition of the domain structure and information of the cucumber BBX proteins. Numbers indicate the amino acid position of the corresponding conserved domains. The green, pink, and yellow rectangles indicate the B-box1, B-box2, and CCT domains, respectively. The scale bar represents 100 amino acids.

**Figure 2 genes-13-00658-f002:**
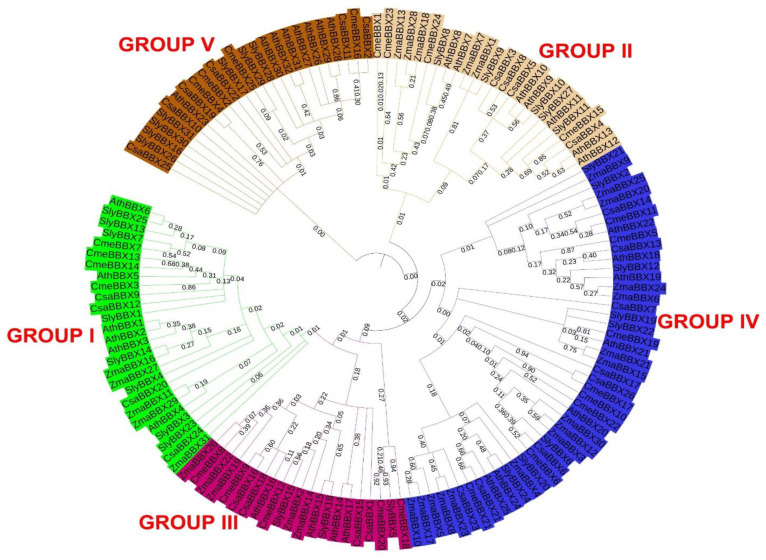
Phylogenetic analysis of BBX proteins from cucumber (*CsaBBX*), Arabidopsis (*AthBBX*), tomato (*SlyBBX*), melon (*CmeBBX*) and maize (*ZmaBBX*). The tree was divided into five clades/groups, which are marked by different colors—green, brown, pink, maroon, and blue, highlighting groups I, II, III, IV, and V, respectively. The bootstrap values are indicated at each node. The phylogenetic tree was performed with MEGA 7.0.26 using the neighbor-joining tree method with 1000 bootstrap replicates. The numbers on the nodes represented the bootstrap replicate.

**Figure 3 genes-13-00658-f003:**
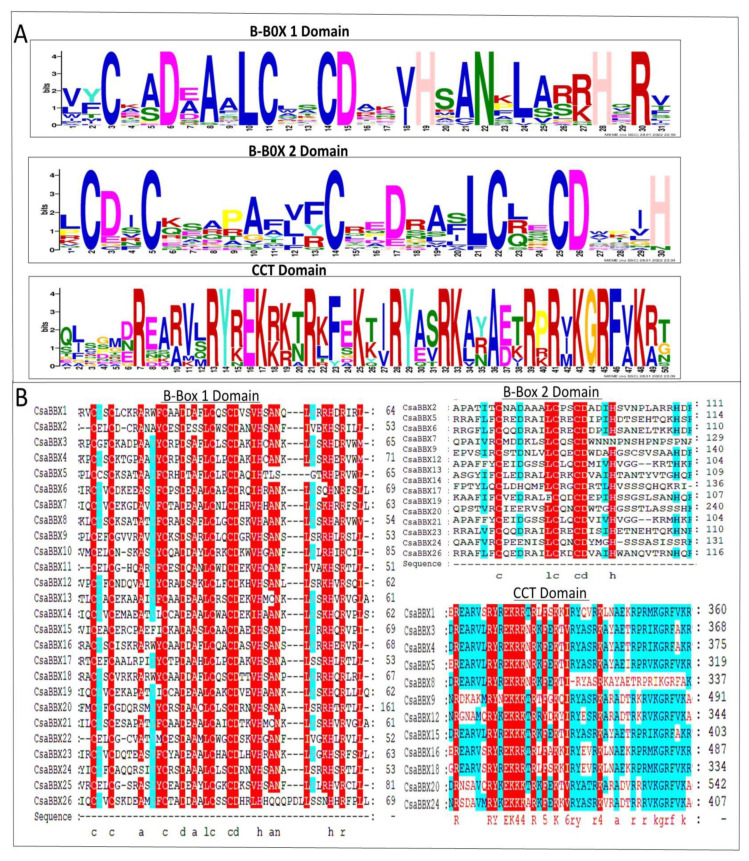
Domain composition of *CsaBBX* proteins. (**A**) The amino acid sequence alignment of the B-box1, B-box2, and CCT domain. The *y*-axis and *x*-axis indicated the conservation rate of each amino acid and the conserved sequences of the domain, respectively. (**B**) Multiple sequence alignments of the B-box1, B-box2, and CCT domains are shown. The identical conserved amino acids are represented by red and light-blue shading.

**Figure 4 genes-13-00658-f004:**
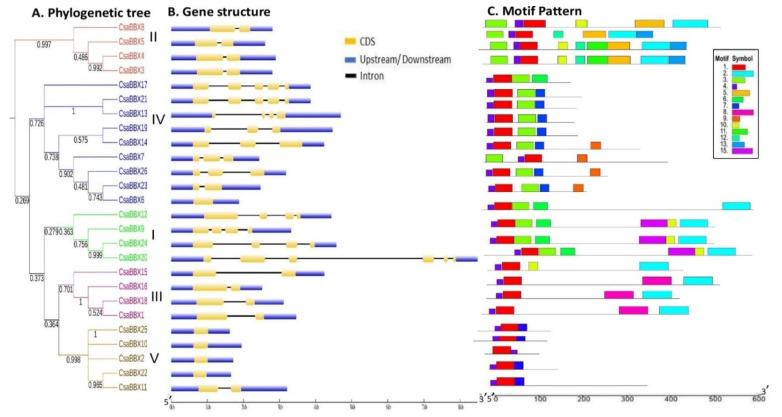
Phylogenetic relationship, gene structure, and architecture of the conserved protein motifs in *CsaBBXs*. (**A**) The phylogenetic tree was constructed based on the full-length sequences of *CsaBBX* proteins. (**B**) The exon-intron structure of *CsaBBXs*. Blue boxes indicate upstream/downstream regions, yellow boxes indicate exons or CDS, and black lines indicate introns. (**C**) The motifs’ compositions. The motifs, numbered 1–15, are displayed in different colored boxes. The sequence information for each motif is provided in additional Appendix A.

**Figure 5 genes-13-00658-f005:**
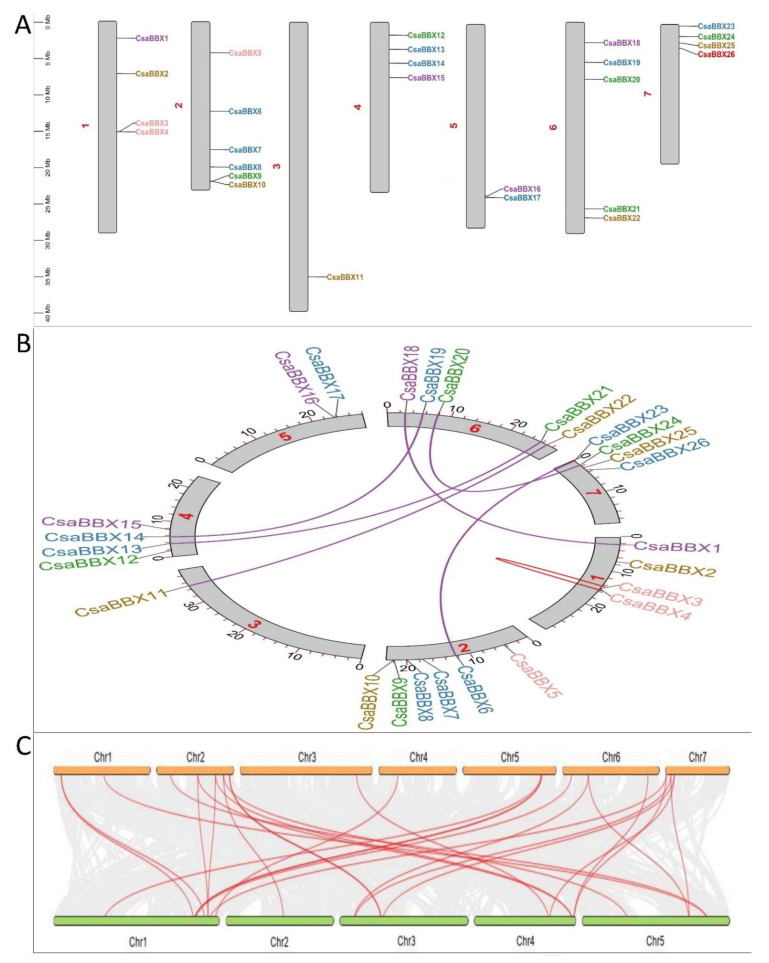
Chromosome distribution analysis of *CsaBBX* genes in cucumber, homologous genes pairs, and synteny analysis. (**A**) Positions of *CsaBBX* genes family members on cucumber chromosomes. (**B**) Circos of duplicated gene pairs; the pink line represents the segmentally duplicated *CsaBBX* genes in cucumber, while the red curved line represents tandem duplication, and the chromosome numbers are labeled in red color. (**C**) Syntenic relationships of BBX gene family in cucumber and Arabidopsis (orange pillar represents *Cucumis sativus* L.; green pillar represents *Arabidopsis thaliana*).

**Figure 6 genes-13-00658-f006:**
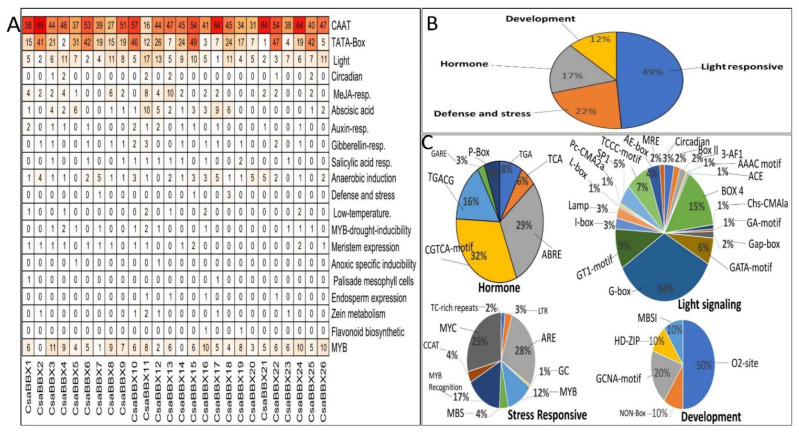
*Cis*-regulatory elements in the promoters of 26 cucumber *CsaBBX* genes. (**A**) The number of various *cis*-regulatory elements in the promoters of each cucumber *CsaBBX* gene. The numbers in boxes are the number of cis-elements in each *CsaBBX* gene. (**B**) The relative proportions of different cis-regulatory elements related to light, hormone, defense stress, and development are indicated by the pie chart. (**C**) The relative proportions of different cis-elements in each category in the promoters of cucumber *CsaBBX* genes are indicated by the pie chart.

**Figure 7 genes-13-00658-f007:**
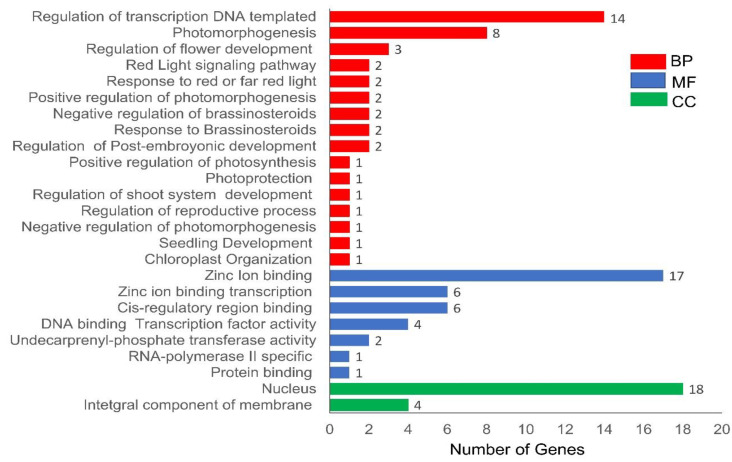
Predicted biological process (BP), molecular functions (MF), and cellular components (CC) of *CsaBBX* genes.

**Figure 8 genes-13-00658-f008:**
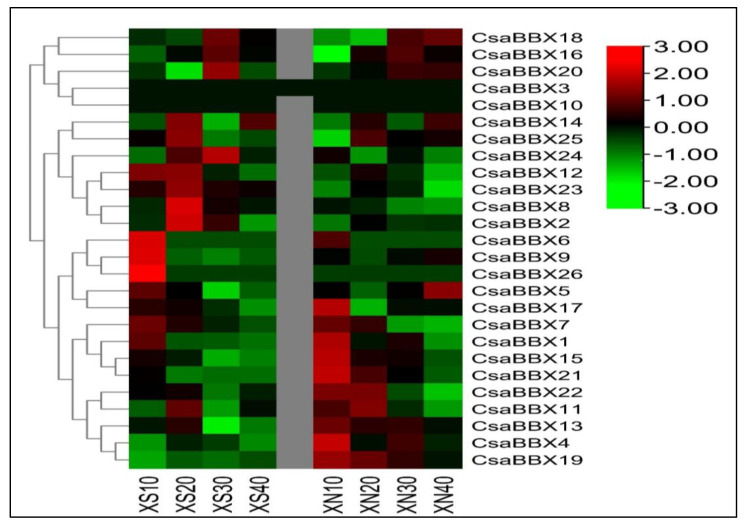
Relative expression of *CsaBBXs* in “XIS” cucumber endocarp fruit. The expression levels of *CsaBBX* in the endocarp of “XIS” cucumber under short-day and day-neutral conditions. XS10-40 and XN10-40 = “XIS” cucumber under short-day and day-neutral conditions at 10 to 40 days after flowering, respectively.

**Figure 9 genes-13-00658-f009:**
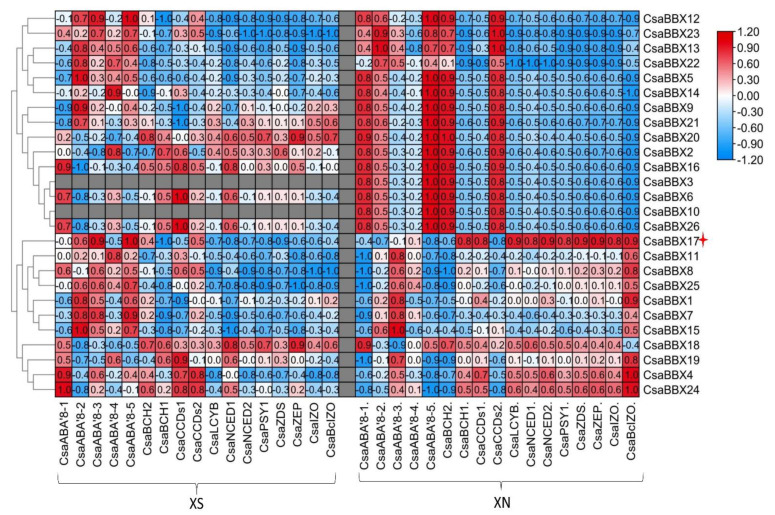
Heatmap of correlation coefficients between *CsaBBX* and carotenoid biosynthesis genes in the endocarp of “XIS” Xishuangbanna cucumber under different photoperiod conditions. The values are correlation coefficients, the pink color on the key shows positive correlations, with higher positive correlations being closer to positive one (+1), while green shows negative correlations, with significantly lower correlations being closer to negative one (−1), XS—short day, XN—day-neutral.

**Figure 10 genes-13-00658-f010:**
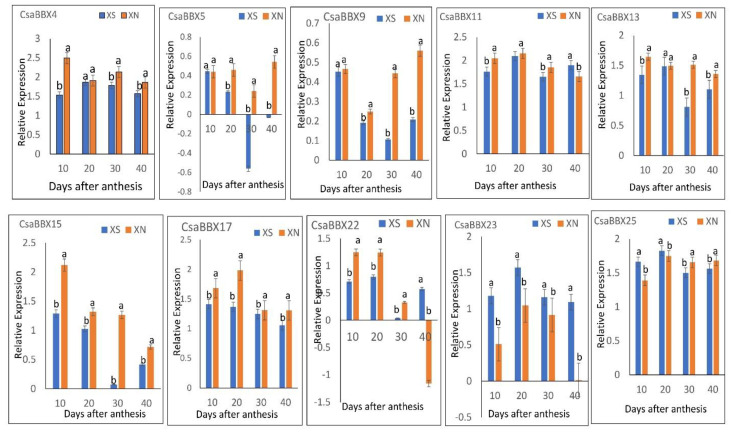
The expression profile of 10 *CsaBBX* in the endocarp of orange-fleshed Xishuangbanna cucumber under short-day (XS) and day-neutral (XN), analyzed by qRT-PCR. The lowercase letters above the bar chart indicate significant differences per data point determined by one-way ANOVA and different letters represent significant differences (*p* < 0.05). Error bars represent standard deviations of the means of triplicates (*n* = 3).

**Table 1 genes-13-00658-t001:** Detailed information of CSaBBX gene family members in cucumber.

				Location									
Gene Name	Accession Number	Gene ID	Chr.	Start	End	CDS/bp	AA	PI	MW (kDa)	I	AI	G	Localization Predicted
*CsaBBX1*	KGN63817	*Csa1G023050*	1	2339530	2339655	1486	360	5.25	41631	51.6	64.97	−0.99	chlo: 5, nucl: 4, mito: 4
*CsaBBX2*	KGN64618	*Csa1G071810*	1	7206080	7206187	517	124	5.35	13749	51.88	71.53	−0.44	nucl: 7, chlo: 4, extra: 2
*CsaBBX3*	KGN65417	*Csa1G420310*	1	15201130	15201268	1516	368	5.77	40463	40.26	69.13	−0.36	cyto: 9, chlo: 4
*CsaBBX4*	KGN65418	*Csa1G420320*	1	15209944	15210082	1601	375	5.79	40992	42.5	69.89	−0.3	chlo: 11, cyto: 2
*CsaBBX5*	KGN61122	*Csa2G057080*	2	4272988	4273095	1291	319	8.2	35158	47.38	66.68	−0.4	chlo: 12, nucl: 1
*CsaBBX6*	KGN61840	*Csa2G250430*	2	12300987	12301125	681	180	6.64	19762	66.9	71.61	−0.3	nucl: 6, chlo: 3, cyto: 3, extra: 1
*CsaBBX7*	KGN62650	*Csa2G365080*	2	17594757	17594895	104	306	6.65	32666	53.09	71.37	−0.22	cyto: 5, chlo: 4, nucl: 4
*CsaBBX8*	KGN63011	*Csa2G383330*	2	19978464	19978602	1499	337	6.08	36934	45.36	68.93	−0.05	chlo: 5, cyto: 4, mito: 3, nucl: 1
*CsaBBX9*	KGN63275	*Csa2G423550*	2	21951054	21951195	1673	396	5.43	43904	52.33	57.65	−0.62	nucl: 12, cyto: 1
*CsaBBX10*	KGN46619	*Csa2G423560*	2	21954107	21954211	751	133	8.4	14655	28.85	79.92	−0.04	chlo: 5, cyto: 4, mito: 3, nucl: 1
*CsaBBX11*	KGN59860	*Csa3G850640*	3	35026996	35027100	1654	274	4.31	30151	60.43	53.72	−0.83	chlo: 10, nucl: 3
*CsaBBX12*	KGN53028	*Csa4G011750*	4	1767965	1768095	2043	491	5.5	54364	41.47	68.51	−0.55	nucl: 9, cyto: 4
*CsaBBX13*	KGN53325	*Csa4G047370*	4	3728156	3729702	1461	186	6.44	20762	51.03	72.37	−0.61	cyto: 11, chlo: 3
*CsaBBX14*	KGN53580	*Csa4G083550*	4	5640054	5641138	1516	297	5.31	32297	57.96	66.4	−0.38	nucl: 14
*CsaBBX15*	KGN53769	*Csa4G124910*	4	7615698	7615836	1228	344	5.24	38471	44.78	57.86	−0.9	nucl: 4, cyto: 4, mito: 3, cysk: 2
*CsaBBX16*	KGN52081	*Csa5G609670*	5	23609342	23609477	1212	403	5.55	45466	47.61	63.27	−0.61	nucl: 10, mito: 2, chlo: 1
*CsaBBX17*	KGN52118	*Csa5G610520*	5	23808456	23808597	1464	487	6.81	54842	63.27	65.66	−0.82	nucl: 13
*CsaBBX18*	KGN45963	*Csa6G039540*	6	2813042	2813177	1302	334	8.33	38396	51.69	74.14	−0.29	nucl: 5, mito: 4, chlo: 3, cyto: 2
*CsaBBX19*	KGN46304	*Csa6G081450*	6	5519827	5519962	1888	237	4.89	26092	41.52	64.43	−0.44	chlo: 7, nucl: 6
*CsaBBX20*	KGN46619	*Csa6G113560*	6	7852876	7853011	1629	542	5.52	60047	58.56	74.23	−0.57	nucl: 4, e.r.: 3, mito: 2, plas: 2, vacu: 2
*CsaBBX21*	KGN48894	*Csa6G505230*	6	25655601	25655890	1224	168	6.59	18863	55.67	74.23	−0.57	cyto: 13
*CsaBBX22*	KGN49170	*Csa6G516780*	6	26902285	26902397	461	123	5.89	13124	50.88	65.77	−0.23	cyto: 7, extr: 6
*CsaBBX23*	KGN43167	*Csa7G004690*	7	236770	236908	1032	222	5.88	24169	56.17	64.77	−0.41	nucl: 11, chlo: 1, cyto: 1
*CsaBBX24*	KGN43392	*Csa7G031530*	7	1670263	1670398	1224	407	5.42	44556	58.02	60.96	−0.52	nucl: 12.5, cyto_nucl: 7
*CsaBBX25*	KGN43541	*Csa7G044810*	7	2580621	2580725	426	132	6.4	14627	37.85	75.3	−0.24	chlo: 9, nucl: 3, cyto: 1
*CsaBBX26*	KGN43667	*Csa7G056460*	7	3310645	3311022	999	332	5.11	36652	43.78	86.66	−0.03	e.r.: 5, plas: 3, cyto: 2, nucl: 1

Chr—chromosome number; CDS/bp—coding sequence or base-pair length; AA—amino acid size; PI—isoelectric point; MW—molecular weight; I—instability index; AI—Aliphatic Index; G—GRAVY; nucl, nucleus; mito, mitochondria; chlo, chloroplast; cyto, cytoplasm; extr, extracellular; cysk, cytoskeleton; vacu, vacuole; e.r, endoplasmic reticulumn; plas, plasma membrane.

## Data Availability

The Arabidopsis, maize, tomato, melon, and cucumber BBX protein sequences were downloaded from the *Arabidopsis thaliana* information source (TAIR) database (https://www.arabidopsis.org/, 1 November 2021), ( maize database (https://download.maizegdb.org/, accessed on 2 November 2021), tomato genome (http://solgenomics.net/ftp/tomato_genome, accessed on 2 November 2021), melon genome (http://cucurbitgenomics.org/organism/18, accessed on 2 November 2021) and cucumber genome (http://cucurbitgenomics.org/organism/2, accessed on 2 November 2021), respectively. RNA-seq photoperiod treatment on “XIS” cucumber (PRJNA782229 in the SRA database (https://www.ncbi.nlm.nih.gov/sra/PRJNA782229, accessed on 1 December 2021).

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
