# Peer review of "Genome-Wide Identification of the B-Box Gene Family and Expression Analysis Suggests Their Potential Role in Photoperiod-Mediated β-Carotene Accumulation in the Endocarp of Cucumber (Cucumis sativus L.) Fruit"

_genes, 2022, doi:10.3390/genes13040658_

Round 1

Reviewer 1 Report

In my opinion Authors present important research results of CsaBBX genes  identified in cucumber. Authors studied their phylogenetic relationship, expression, gene organization and response to changes in light longevity. Study is generally well planned and performed. Conclusions are supported by research results. The description of Materials and Methods together with the results presentation should be improved. In my opinion after major corrections provided below article could be suitable for publication.

Citations should be in the [] bracket not in the ()- correct in the entire text.  Some citation numbers are in bold- correct it in the entire text.

Line 195- add following information: method of RNA isolation, amount of RNA taken for preparaing cDNA libraries, volume and concentration of  libraries.

Section 2.9

Concentration of RNA taken for experiment, how the quality of RNA was assessed?

Provide the name of reference gene- was the stability of reference gene expression tested ? Provide citation of previous use this reference gene.

Add details of PCR reaction. Also name and manufacturer of RT-PCR machin is necessary.

Provide name, sequences and accession numbers of reference gene primers- in Table S4 (additional Text File 1). Also the size of PCR products (tested and reference)  could be presented in Table S4.

Authors may add the accession numbers to proteins in Text file S2.

Line 320. Additional Table 1- this file is named not Table S1, but the Additional file 1-correct the name. Maybe write in the text : Additional file 1: Table S1. The same mode for other Tables S2-S4 present in Additional file 1, for example instead Additional  file 1 could be Additional file 1: Tables S1-S4. It will be easier to know what is inside.

Fig 4B- the nuance/shade of blue in figure 4B  and in description Upstream/Downstream is different, correct it to be the same as on the figure.

Fig. S1- there is a problem with readability, the color nuances are partially too related, maybe instead of color shade/nuance Authors could just number these cis-active motifs for example by Roman numbers; I, II, II, IV etc. It could be much more clear to find them.

Additional File 2 could be named  Additional File 2: Fig S1  it will be much more clear to localize it.

Table S4- units of beta carotene concentration should be presented for example µg/g dry weight or fresh weight. State it also clearly in HPLC method description.

Author Response

We are very grateful for taking your time to review our manuscript. Following your comments and suggestions, we have tried our level best to address the issues in order to improve the content of the manuscript. Please see the attachment for detailed responses. Thank you very much in advance for your support.

Reviewer 2 Report

Dear Author,

You can find my post review report attached. I have stated the suggestions and some minor corrections that I consider necessary there.

Kind Regards

Title

OK

 Abstract

Line 21, B-Box …

Line 22 and 24, BBX,

Line 26, CsaBBXs……

……… All genes mentioned in the text should be written in italics. These should be corrected throughout the manuscript.

Line 31, “Arabidopsis” should be Arabidopsis thaliana (L.) Heynh.

 Keywords

In general, it should be more correct to avoid the use of the same terms used within the title. It would be appropriate to review the keywords and add the ones that are not included in the title. Just a suggestion, CsaBBXs genes, can be added.

 Introduction

In all manuscripts, references are indicated with "(x)"

…In the text, reference numbers should be placed in square brackets [ ], and placed before the punctuation; for example [1], [1–3] or [1,3]. For embedded citations in the text with pagination, use both parentheses and brackets to indicate the reference number and page numbers; for example [5] (p. 10). or [6] (pp. 101–105)……

Line 71, “in vitro” …. should be written in italics…

 Material Methods

Line 169, “(C. s Xishuangbannesis)” correct …. Cucumis sativus L. var. xishuangbannesis Qi et Yuan

Line 186, ….. all other environmental factors were the same …. (Brief information about these factors should be given. eg temperature etc.)

Line 202, subtitle… “2.8β-. carotene extraction and HPLC analysis”….. correct

2.8. β-carotene extraction and HPLC analysis

Line 203, …. the methods described by (39). ……….. sentence seems to be incomplete…

……described by Alagoz et al (39).

Line 205, “2 mL” …. “2 ml”

Line 208, “revolutions per minute” not necessary, rpm is enough

Line 218, “…….Sigma-Aldrich Co. Ltd, Shanghai, China).” Where do the parentheses start?

 Results

OK

 Discussion

Line 517, (Fig S2)???

Tables

For the same word in the table and in its legends, there is a use of lowercase letters in some places and capitalization in others. For example, Chlo – chlo; Cyto – cyto; Index – index …. A standard usage of them would be fine.

 Figures

OK

 References

OK

Author Response

(The authors gave the same response as above.)

Round 2

Reviewer 1 Report

In my opinion Authors included all corrections, that were suggested in my review, now I have no comments to the manuscript and it could be published. 

Author Response

We are very grateful for taking your time to review our manuscript. Following your recommendation for the publication of our manuscript, we owe you great gratitude for your support as we look forward to its further processing. Thank you very much.